# FedDdrl: Federated Double Deep Reinforcement Learning for Heterogeneous IoT with Adaptive Early Client Termination and Local Epoch Adjustment

**DOI:** 10.3390/s23052494

**Published:** 2023-02-23

**Authors:** Yi Jie Wong, Mau-Luen Tham, Ban-Hoe Kwan, Yasunori Owada

**Affiliations:** 1Department of Electrical and Electronic Engineering, Lee Kong Chian Faculty of Engineering and Science, Universiti Tunku Abdul Rahman, Kajang 43000, Malaysia; 2Department of Mechatronics and Biomedical Engineering, Lee Kong Chian Faculty of Engineering and Science, Universiti Tunku Abdul Rahman, Kajang 43000, Malaysia; 3Resilient ICT Research Center, Network Research Institute, National Institute of Information and Communications Technology (NICT), Tokyo 184-8795, Japan

**Keywords:** federated learning, client selection, local epoch adjustment, deep reinforcement learning, Internet of Things

## Abstract

Federated learning (FL) is a technique that allows multiple clients to collaboratively train a global model without sharing their sensitive and bandwidth-hungry data. This paper presents a joint early client termination and local epoch adjustment for FL. We consider the challenges of heterogeneous Internet of Things (IoT) environments including non-independent and identically distributed (non-IID) data as well as diverse computing and communication capabilities. The goal is to strike the best tradeoff among three conflicting objectives, namely global model accuracy, training latency and communication cost. We first leverage the balanced-MixUp technique to mitigate the influence of non-IID data on the FL convergence rate. A weighted sum optimization problem is then formulated and solved via our proposed FL double deep reinforcement learning (FedDdrl) framework, which outputs a dual action. The former indicates whether a participating FL client is dropped, whereas the latter specifies how long each remaining client needs to complete its local training task. Simulation results show that FedDdrl outperforms the existing FL scheme in terms of overall tradeoff. Specifically, FedDdrl achieves higher model accuracy by about 4% while incurring 30% less latency and communication costs.

## 1. Introduction

In the Internet of Things (IoT), each device can collect massive amounts of data (i.e., measurements and location information) [1]. It is estimated that IoT devices will generate over 90 zettabytes of data globally by 2025 [2]. These data can be uploaded to a centralized server, where a new model can be retrained or fine-tuned using the collected dataset. This method is called centralized learning (CL) since the data must be centralized at one location for model training. However, privacy concerns make it inconvenient for devices to share potentially sensitive data with a centralized server (or any other party). For example, medical images may contain sensitive and private information about patients [3], which prohibits the collection of such data from multiple healthcare institutions for CL. Additionally, uploading bandwidth-hungry data requires a high communication cost, which is not feasible for most IoT devices with network resource constraints [1,4]. In fact, the data collected by IoT devices could be larger than the model size [4], especially when dealing with image data.

Federated learning (FL) has emerged as one of the promising candidates to address this challenge. FL is a technique that trains an algorithm across multiple edge devices holding individual local datasets without sharing or exchanging them. First, each IoT device (client) uses its locally collected data to train a local model. After training, each IoT device uploads its locally trained models to an FL server for aggregation. A new global model is generated, which is trained using data from all participating clients without actually sharing the sensitive and bandwidth-hungry data. Thus, FL addresses data privacy concerns by training a global model in distributed environments. FL has since been applied in various applications, ranging from mobile keyboard prediction [5] to natural disaster classification [6,7] and medical image segmentation [3]. Using FL, Google trained its mobile keyboard prediction using 600 million sentences from a surprising amount of 1.5 million clients [5]. In addition, Intel released its production-ready and open-source FL (OpenFL) framework [8]. OpenFL is used in the Federated Tumor Segmentation (FeTS) initiative, which is a program participated in by 56 clinical sites around the globe to train tumor segmentation models via FL. Experiments show that FL models can reach 99% of CL model without sharing the sensitive data [3]. These large-scale real-life applications proved the huge economic value of FL.

FL coupled with the IoT has huge potential for real-world application. For instance, research works [9,10,11] combined FL with industrial IoT (IIoT), creating an industrial-grade hierarchical FL framework. Hierarchical FL is a three-layer architecture FL framework composed of clients, edge servers and a cloud server. Regular FL is performed between the edge server and its corresponding client device. Upon model aggregation at the edge servers, the aggregated models are then uploaded to the cloud for global model aggregation. Experiments show hierarchical FL to be superior to a regular FL, with lower training latency and better convergence [12]. This is because the model aggregation at the client edge before global model aggregation can significantly reduce the training divergence. Due to the robustness of hierarchical FL, it has also been exploited to empower digital twins [10,13]. In this study, we only focus on regular FL, which is the fundamental building block for any sophisticated FL framework.

Despite huge potential, FL still faces several challenges from practical implementation: (1) model convergence in the presence of a non-independent and identically distributed (non-IID) dataset, (2) computing efficiency and (3) communication efficiency [14,15]. First, data is usually not uniformly distributed across IoT devices. Realistically, each IoT device has a unique data distribution and can be considered non-IID, whereas the global population (if the data is centralized) would be IID. According to [16], the earth mover’s distance $\left(\mathrm{EMD}\right)$ between the local client data distribution and the global population is the main reason the FL model diverges from the global optima solution. This is also termed weight divergence between FL and CL models, which greatly reduces the convergence rate of FL models [16]. Additionally, the heterogeneity of computing and communication resources in IoT networks hinders resource utilization efficiency. In most studies, except [17,18], the local epoch number is set to be the same for all client devices disregarding their computing constraints. As a result, devices with stronger computing power often have to wait for the straggler devices to complete their training, which drastically increases the overall training latency. Moreover, some clients may not have access to high-speed networks, making local model uploading slow or unrealistic.

Many previous studies aimed to tackle the three challenges from different viewpoints. However, optimizing one of the objectives might deteriorate the other objectives [14]. Deep reinforcement learning (DRL) has recently been exploited for FL resource optimization. However, to the best of our knowledge, none of the DRL-based FL frameworks allows dynamic local epoch adjustment. Past studies [9,10,14,19,20,21] only exploited DRL to select clients that fulfil the resource constraints without introducing a tuning mechanism to adjust the local training epoch for clients with limited computing power.

To this end, we present federated double deep reinforcement learning (FedDdrl). FedDdrl exploits the double DRL (DDRL) framework, which uses two DRL agents to find the optimal client selection and local epoch adjustment policies. Our objective was to maximize the global model’s accuracy while minimizing the FL system’s training latency and communication cost. We first formulated the FL protocol as a Markov Decision Process (MDP). We then adopted two DRLs based on Value Decomposition Networks (VDNs) as the policy networks. To speed up the convergence speed, we adopted the recently proposed balanced-MixUp [22] augmentation technique to mitigate weight divergence. Simulation results showed that our FedDdrl algorithm improved model accuracy with lower training latency and communication cost. Note that FL facilitates edge computing, which is one of the goals of the ASEAN IVO project titled “Context-Aware Disaster Mitigation using Mobile Edge Computing and Wireless Mesh Network”.

We summarize our contributions as follows.

We modeled the FL system as an MDP. Then, we proposed to use a DDRL framework for adaptive early client termination and local epoch adjustment, to maximize the global model accuracy while minimizing the training latency and communication costs.We demonstrated our proposed algorithm in a non-IID setting on MNIST, CIFAR-10 and CrisisIBD datasets. We showed that our solution could outperform existing methods in terms of global model accuracy with shorter training latency and lower communication costs.We explored the influence of balanced-MixUp in the FL system. In most settings, balanced-MixUp could mitigate weight divergence and improve convergence speed.

The rest of the paper is organized as follows. Section 2 describes the related work. Section 3 discusses the system model and problem formulation. Section 4 presents the proposed solution. Section 5 shows the experimental setup, followed by the simulation results and discussion. Section 6 concludes the paper and outlines future research directions.

## 2. Related Work

This section reviews existing works on FL and DRL-based FL to provide insights into the current trend in FL. Then, we discuss the limitation of each algorithm. Lastly, we also elaborate on the weight divergence problem in FL, which is a common problem faced by all FL algorithms.

### 2.1. Federated Learning and Deep Reinforcement Learning

Some of the commonly used FL algorithms include FedAvg [4], FedProx [23] and FedNova [17]. FedAvg was the first practical implementation of FL. In each communication round, the server sends the global model to N randomly selected clients. Each client trains the model using its local dataset. Then, each client uploads its locally trained model to the server, where the server averages the received local models’ weights as the new global model. It has since become the de facto approach for FL and is widely used in various applications [3,5,6,7]. FedProx presents a reparameterization of FedAvg by introducing an additional $L_{2}$ regularization term in the local objective function. The regularization term limits the distance between the local and global models, preventing local updates from diverging from global optima. A hyperparameter μ controls the weight of the regularization term. Overall, the modification can be easily performed on the existing FedAvg algorithm while improving model accuracy on non-IID datasets. However, it introduces additional computing overhead, leading to longer training latency. On the other hand, FedNova improves FedAvg in the aggregation stage. It allows each n∈N client to conduct a different number of local steps. This allows clients with higher computing resources to conduct more training while waiting for others to complete training. To ensure that the global updates are not biased, each local update is normalized and scaled according to the number of local steps conducted before they are averaged into the new global model. FedNova introduces negligible computation overhead compared to FedAvg while handling computing resources heterogeneity in FL systems. However, all the above algorithms are limited to handling statistical heterogeneity (non-IID dataset) and computing resources heterogeneity. Other heuristic algorithms have been proposed to optimize client selection in FL systems with heterogeneous network and/or energy resources [24,25]. However, heuristic solutions could only deliver sub-optimal performance since they often rely on qualitative analysis without exploring the optimal performance [14].

DRL has been widely applied to solve optimization problems involving complex sequential decision-making, such as playing Atari games [26], multiplayer games [27] and chess [28]. Since an FL procedure can be modelled as an MDP, it can also be optimized using DRL. FAVOR is one of the first research works to optimize FL using DRL [21]. They observed an implicit connection between the distribution of a local client dataset and the model weights trained on those data. Using the model weights collected from each participating client, a DRL agent can learn to select suitable clients for the next round of training. After proving DRL success in FL optimization, multiple studies [9,10,14,19,20,29] have exploited DRL in FL resource allocation problems. For instance, [9,10,14,19] used DRL to jointly optimize computing and network resources in an FL framework while retaining the global model’s accuracy. These studies employed a DRL-based client selection policy or early client termination policy. Such a policy is responsible for selecting the best subset of clients for each round of training by optimizing the tradeoff between model accuracy and resource allocation. On the other hand, [29] optimized only the network resources by quantizing the model weights before uploading them to the FL server. However, none of the DRL-based FL frameworks described above allow dynamic local epoch adjustment. These frameworks fixed the same local epochs for all clients, disregarding their computing cost and training latency. With dynamic local epoch adjustment, clients with higher computing resources can conduct more training epochs. On the contrary, clients with limited computing power can train with fewer epochs.

Table 1 summarizes the key features of the aforementioned FL and DRL-based FL algorithms. In short, we noticed an ongoing trend of utilizing DRL to optimize the computing and network resources in the FL framework. Most of them relied only on client selection or early client termination techniques. As a result, such a method often rejects clients with limited computing power to prevent these devices from dragging the overall FL training latency. Even when such devices are selected, those with stronger computing power will finish training earlier and remain idle while waiting for the slower ones to complete training. However, these devices may contain crucial training data that is essential for FL convergence. Ideally, these devices should participate in FL training but with a lower local training epoch and vice versa. To the best of our knowledge, no DRL-based FL algorithms adopt DRL for automated local epoch adjustment. On the other hand, existing FL algorithms such as FedNova rely on manual adjustment to set devices with stronger computing power with a higher local epoch. Hence, an exciting potential exists for incorporating DRL-based dynamic local epoch adjustments for automated calibration.

### 2.2. Weight Divergence in Federated Learning

Weight divergence is the difference between FL model weights $w_{t}^{FL}$ and CL model weights $w_{t}^{CL}$. An ideal level of weight divergence in FL could exploit the rich decentralized data, resulting in a better performance. For instance, FL outperforms its CL counterpart in various applications, including drug discovery [30], disaster classification [7] and autonomous driving object detection [31]. However, in extreme non-IID cases where the local client data distribution $p^{k}$ is far from the global data distribution p, the highly diverged local weight updates could lead to bad aggregated solutions which are far from the global optimum solution. This is especially true in IoT networks, where each IoT device has a unique data distribution and can be considered non-IID [1]. According to [16], the main source of weight divergence is the earth mover’s distance $\left(\mathrm{EMD}\right)$ between $p^{k}$ and p, denoted as $\sum_{i=1}^{n_{c}}∥p^{k}\left(y=i\right)-p\left(y=i\right)∥$, where $n_{c}$ denotes the total number of classes. In general, weight divergence is inevitable since $p^{k}$ and p are almost guaranteed to be different in a real-life setting. Figure 1 shows an example of weight divergence between FedAvg and CL models.

Generally, only a fraction of the total clients is selected for training per communication round, t, to reduce the total communication cost in FL. Let C denote the total number of participating clients per round. When C is low, it is difficult to ensure the sampled data resemble the global data distribution. This also leads to high EMD between $p^{k}$ and p, which again contributes to the divergence of $w_{t}^{FL}$ from $w_{t}^{CL}$. Let $A_{t}$ be the accuracy of the global model at communication round t∈T. Figure 2 shows the accuracy curve of FedAvg models trained on a non-IID CIFAR-10 dataset using C=5 and C=10. First, the average accuracy of the global model after t=15 communication round was 62.73% and 72.79% for C=5 and C=10, respectively. Additionally, the fluctuation and standard deviation of the accuracy curve were higher when C=5 as compared to C=10. It is shown that FedAvg (or FL in general) had inferior performance when the number of participating clients per round is low.

Recent studies have contributed various solutions to mitigate weight divergence. For instance, the FedProx [23] mentioned earlier adds a regularization term to the local subproblem to prevent the local updates from diverging away from the global FL model. This method, in turn, hopes to ensure the aggregated global FL model weights $w_{t}^{FL}$ are close to $w_{t}^{CL}$. Albeit effective, FedProx requires higher computing costs and a longer training time [32]. On the other hand, [16] proposed partial global sharing of local data to reduce EMD between client data distribution and the global populations. However, this induces high communication costs for data sharing and raises privacy concerns.

Meanwhile, methods that employ adaptive client selection or early client termination (i.e., FedMarl) aim to tackle weight divergence via careful client selection. For each communication round, FedMarl will only select a subset of the C clients for training. Ideally, only the selected clients are useful for training, while the rest are not. Effectively, this means that C is not constant for each communication round t. However, a lower C may lead to less steady convergence based on Figure 2. Thus, FedMarl is expected to handle the careful dropping of clients considering the EMD between global and local populations while taking care of other optimizing objectives, such as the training latency and communication cost of each client. Dropping the wrong clients may lead to large weight divergence, as shown in Figure 3.

## 3. System Model and Problem Formulation

In this section, we present the system model for our FL system and discuss the problem formulation. Our commonly used symbols are listed in Table 2 for ease of reference.

### 3.1. System Model

We considered an FL system with K number of client devices. At communication round t∈T, N number of clients were randomly selected from the K number of clients. Each communication round consisted of four phases, which are: (1) model broadcasting, (2) probing training, (3) client dropping and (4) completion of training.
Model broadcasting: If t=1, the FL server will initialize a global model, whereas at t≥2, the FL server will collect the client models trained at round t−1 and aggregate them into a new global model. Then, the FL server will broadcast the global model to N randomly selected clients.Probing training: Each selected client n∈N will perform one epoch of local training called probing training. The purpose of probing training is to acquire the metadata of each client. The metadata consist of the client’s states, which will be fed to the DRL agents for adaptive early client termination and local epoch adjustment. The details of the client states will be defined later together with the specification of the DRL agents. After probing training, each client will upload its metadata to the server and proceed to the next phase.Early client termination: Based on the collected client states, the DRL agents at the FL server will drop non-essential clients to reduce total latency $H_{t}$ and total communication cost $B_{t}$ for round t. The decision made by DRL agents will be sent to each client.Completion of training: Only the remaining C clients that are not dropped by the DRL agent will resume training. Each client n will complete the remaining local training until $E_{n}^{t}$ epochs are reached. Each locally trained model will be uploaded to the FL server for model aggregation.

Let $H_{t,n}^{p}$ denote the probing training latency for client n∈N at round t∈T. Let $H_{t,n}^{c}$ be the complete local training latency for client n at round t, while $H_{t,n}^{u}$ denotes the time taken for client n to upload its local model to the FL server. Let $\phi_{n}^{t}\in \left\{0,1\right\}$ denote if client n is selected by the DRL agent to complete full local training. The total processing latency $H_{t}$ and the total communication cost $B_{t}$ at communication round t can be expressed by Equations (1) and (2):(1)$$H_{t}=\underset{1\leq n\leq N}{\max}\left(H_{t,n}^{c}+H_{t,n}^{u}\right)a_{n}^{t}$$
(2)$$B_{t}=\overset{N}{\underset{n=1}{\sum }}B_{n}^{t}\phi_{n}^{t}$$

Figure 4 depicts the system model for the proposed FL protocol. Note that the model broadcasting latency $H_{t,n}^{b}$ was not included into $H_{t}$ since it is not part of the FL optimization problem. Additionally, the time latency to upload client metadata to the server, $H_{t,n}^{m}$, was ignored. This is because the metadata file size was only 278 bytes, while even a lightweight MobileNetV2 file is 24.5 megabytes. Thus, $H_{t,n}^{m}$ is negligible since $H_{t,n}^{m}\ll H_{t,n}^{u}$.

### 3.2. Problem Formulation

Our objective was to maximize the cumulative $A_{t}$ improvement while minimizing the $H_{t}$ and $B_{t}$. Let $\phi_{t}=\left[\phi_{n}^{t}\right]$ and $E_{t}=\left[E_{n}^{t}\right]$ be a T×N matrix for client termination and local epoch adjustment decided by the DRL agents, respectively. We formulated the problem as a weighted sum optimization problem, as formulated below:(3)$$\underset{\phi_{t},E_{t}}{\max}\;E\left[\overset{T}{\underset{t=1}{\sum }}w_{1}\left[U\left(A_{t}\right)-U\left(A_{t-1}\right)\right]-\left(w_{2}B_{t}+w_{3}H_{t}\right)\right]$$
where $w_{1}$, $w_{2}$ and $w_{3}$ are the weights to control the importance of each objective. To ensure $A_{t}$ can improve even if it is small near the end of the FL process, a utility function denoted as U(·) was used to reshape the $A_{t}$ of the global model. In FedMarl, $U\left(A_{t}\right)$ is defined in Equation (4):(4)$$U\left(A_{t}\right)=\frac{20}{1+e^{0.35\left(1-A_{t}\right)}}-10$$

One problem with the original $U(A_{t})$ is that it only tells us the transformed value of $A_{t}$. The entire $w_{1}\left[U\left(A_{t}\right)-U\left(A_{t-1}\right)\right]$ can be reparametrized into a single $w_{1}U\left(\Delta A_{t}\right)$ expression, which could directly tell us the gain/penalty for $\Delta A_{t}$. First, the $U\left(A_{t}\right)$ equation is simplified in the given range $0\leq A_{t}\leq 1$ since $A_{t}$ is bounded between 0 and 100%. In this range, $U\left(A_{t}\right)$ can be approximated as a straight line, as shown in Figure 5.

To approximate $U\left(A_{t}\right)$ as a straight line in the given range, the gradient and y-intercept of the graph are required. The gradient is denoted as $U^{\prime }\left(A_{t}\right)$, which is the first derivative of the $U\left(A_{t}\right)$ function. $U^{\prime }\left(A_{t}\right)$ can be written as in Equation (5):(5)$$U^{\prime }\left(A_{t}\right)=\frac{7e^{0.35\left(1-A_{t}\right)}}{{\left(1+e^{0.35\left(1-A_{t}\right)}\right)}^{2}}$$

The mean of the gradient, $\bar{U^{\prime }\left(A_{t}\right)}$, within the range can be formulated as in Equation (6):(6)$$\begin{array}{l} \bar{U^{\prime }\left(A_{t}\right)}\\ =\frac{1}{1-0}\int_{0}^{1}U^{\prime }\left(A_{t}\right)dt\\ =\int_{0}^{1}\frac{7e^{0.35\left(1-A_{t}\right)}}{{\left(1+e^{0.35\left(1-A_{t}\right)}\right)}^{2}}\;dt\\ =1.732 \end{array}$$

The y-intercept of $U\left(A_{t}\right)$, denoted as $U\left(A_{t}=0\right)$, can be written as:(7)$$\begin{array}{l} U\left(A_{t}=0\right)\\ =\frac{20}{1+e^{0.35\left(1-0\right)}}-10\\ =-1.732 \end{array}$$

Hence, $U\left(A_{t}\right)$ can be simplified into:(8)$$\begin{array}{l} \left(A_{t}\right)\\ =\bar{U^{\prime }\left(A_{t}\right)}\;A_{t}+U\left(A_{t}=0\right)\\ =1.732\;A_{t}-1.732 \end{array}$$

Thus, $U\left(\Delta A_{t}\right)$ can be defined as:(9)$$\begin{array}{l} U\left(\Delta A_{t}\right)\\ ≅U\left(A_{t}\right)-U\left(A_{t-1}\right)\\ =\left(1.732\;A_{t}-1.732\right)-\left(1.732\;A_{t-1}-1.732\right)\\ =1.732\left(\;A_{t}-A_{t-1}\right)\\ =1.732\;\Delta A_{t} \end{array}$$

$U\left(\Delta A_{t}\right)$ is more analyzable than $U\left(A_{t}\right)-U\left(A_{t-1}\right)$ since the two expressions have been collapsed into one equation. At this end, we can define our optimization problem as:(10a)$$\underset{\phi_{t},E_{t}}{\max}\;E\left[\overset{T}{\underset{t=1}{\sum }}w_{1}U\left(\Delta A_{t}\right)-\left(w_{2}B_{t}+w_{3}H_{t}\right)\right]$$
(10b)$$s.t.\;w_{1},w_{2},w_{3}>0$$
(10c)$$E\left(w_{3}\overset{T}{\underset{t=1}{\sum }}H_{t}\right)>\Omega_{1}E\left(w_{2}\overset{T}{\underset{t=1}{\sum }}B_{t}\right)\;$$
(10d)$$w_{1}U\left(\Delta A_{t}=0.01\right)>\Omega_{2}E\left(w_{2}B_{t}+w_{3}H_{t}\right)$$
(10e)$$E\left(w_{1}\overset{T}{\underset{t=1}{\sum }}U\left(\Delta A_{t}\right)\right)>\Omega_{3}\left(w_{2}\overset{T}{\underset{t=1}{\sum }}B_{t}+w_{3}\overset{T}{\underset{t=1}{\sum }}H_{t}\right)\;$$
where (10b–e) are the constraints for our optimization problem. Constraint (10b) is to make sure the sign of $U\left(\Delta A_{t}\right)$, $B_{t}$ and $H_{t}$ are not inverted. Meanwhile, constraint (10c) is to control the ratio of $\overset{T}{\underset{t=1}{\sum }}B_{t}$ to $\overset{T}{\underset{t=1}{\sum }}H_{t}$. Furthermore, constraint (10d) is to make sure $w_{1}U\left(\Delta A_{t}\right)$ gain will not be outweighed $\left(w_{2}B_{t}+w_{3}H_{t}\right)$ penalties when $\Delta A_{t}$ is as small as 0.01. Lastly, constraint (10e) makes sure $\overset{T}{\underset{t=1}{\sum }}w_{1}U\left(\Delta A_{t}\right)$ is at least $\Omega_{3}$ greater than the penalty terms. Note that (10c-e) are additional constraints that are not imposed on the original FedMarl optimization problem.

In FedMarl, the $w_{1}$, $w_{2}$ and $w_{3}$ are treated as hyperparameters. FL engineers have to manually adjust the weightage of each objective until the desired outcome is achieved. However, the weightage $w_{1}$, $w_{2}$ and $w_{3}$ does not directly translate to the weightage of each objective $\overset{T}{\underset{t=1}{\sum }}U\left(\Delta A_{t}\right)$, $\overset{T}{\underset{t=1}{\sum }}B_{t}$ and $\overset{T}{\underset{t=1}{\sum }}H_{t}$. For instance, the ratio of $w_{2}B_{t}$ to $w_{3}H_{t}$ does not directly equate to the ratio of $w_{2}\overset{T}{\underset{t=1}{\sum }}B_{t}$ to $w_{3}\overset{T}{\underset{t=1}{\sum }}H_{t}$. This is because the values of $H_{t}$ and $B_{t}$ are instantaneous and stochastic, which means that the ratio of $w_{3}H_{t}$ to $w_{2}B_{t}$ at two different t is most likely different. On the other hand, $E\left(\overset{T}{\underset{t=1}{\sum }}H_{t}\right)$ and $E\left(w_{2}\overset{T}{\underset{t=1}{\sum }}B_{t}\right)$ are more consistent. Taking the ratio of these two components is more reliable.

We can find the best $w_{1}$, $w_{2}$ and $w_{3}$ by setting the desired $\Omega_{1}$, $\Omega_{2}$ and $\Omega_{3}$. We set $\Omega_{1}=0.2$, $\Omega_{2}=0.3$ and $\Omega_{3}=1.0$. We needed to run one iteration of FL using FedAvg to get the traces value of $\Delta A_{t}$, $B_{t}$ and $H_{t}$ for t∈T since these values are dependant on the target IoT environment setup. Based on the traces value, we could follow Algorithm 1 to acquire the suitable $w_{1}$, $w_{2}$ and $w_{3}$. In our experiment setup, we found the desired hyperparameters to be ($w_{1}$ = 2.9, $w_{2}$ = 0.1 and $w_{3}$ = 0.2).
**Algorithm 1** Search for the best $w_{1}$, $w_{2}$ and $w_{3}$
1.**Input:**Set $\Omega_{1}=0.2$, $\Omega_{2}=0.3$, $\Omega_{3}=1.0$
2.**Output:**The best $w_{1}$, $w_{2}$, $w_{3}$
3:Run one complete iteration of FedAvg with T=15 communication rounds and record the traces value of $\Delta A_{t}$, $B_{t}$ and $H_{t}$ for t∈T.4:Initialize an empty set $cache=\left\{\right\}$ to store all $\left(\left(w_{1},w_{2},w_{3}\right),R\right)$ that satisfied constraints (10c-e)R is the weighted-sum optimization goal $w_{1}U\left(\Delta A_{t}\right)-\left(w_{2}B_{t}+w_{3}H_{t}\right)$
5:**for** $w_{1}=0,\;0.1,\;\ldots 3.0$ **do**6:
**for** $w_{2}=0,\;0.1,\;\ldots 1.0$ **do**7:

**for**$w_{3}=0,\;0.1,\;\ldots 1.0$ **do**8:


Compute $E\left[w_{1}\overset{T}{\underset{t=1}{\sum }}U\left(\Delta A_{t}\right)\right]$, $E\left[w_{2}\overset{T}{\underset{t=1}{\sum }}B_{t}\right]$, $E\left[w_{3}\overset{T}{\underset{t=1}{\sum }}H_{t}\right]$, $E\left(w_{2}B_{t}+w_{3}H_{t}\right)$ based on the recorded traces value, where we assume $E\left(x\right)≜x$
9:


**if** (10c-e) are satisfied:10:



Compute $R=\overset{T}{\underset{t=1}{\sum }}w_{1}U\left(\Delta A_{t}\right)-\left(w_{2}B_{t}+w_{3}H_{t}\right)$ from the traces value11:



Record $\left(\left(w_{1},w_{2},w_{3}\right),R\right)$ in cache
12:


**end if**13:

**end for**14:
**end for**15.**end for**16:From cache, find out which combination of $\left(w_{1},w_{2},w_{3}\right)$ results in the smallest R. This can be treated as finding the worse-case $\max\;E\left[R\right]$.17:**return**$w_{1},w_{2},w_{3}$

## 4. Proposed Method

We propose FedDdrl, which exploits two DRL policy networks for FL optimization. Specifically, we adopted VDNs as the DRL policy networks for FedDdrl. We elaborate in detail on how we formulated the problem as MDP, including the design of state space, action space and reward of the algorithm. In addition, we also exploited the recently proposed balanced-MixUp to mitigate the impact of weight divergence and speed up the FL convergence speed.

### 4.1. Deep Reinforcement Learning for Federated Learning Optimization

The proposed optimization problem in Equation (10a–e) is a 0–1 Multidimensional Knapsack Problem (MKP). The items to be put in knapsacks are the client devices n with complete training latency $H_{t,n}^{c}$, model uploading latency $H_{t,n}^{u}$, communication cost $B_{n}^{t}$ and data size $D_{n}$. The total capacity of the knapsack equals the total communication cost $B_{t}=\overset{N}{\underset{n}{\sum }}B_{n}^{t}a_{n}^{t}$, where $a_{n}^{t}$ is the binary indicator of item (client) n. When $a_{n}^{t}$ is set to 1, item (client) n is selected. Otherwise, $a_{n}^{t}$ is set to 0. The total weight of the knapsack has a lower bound which has to fulfil the minimum requirement of accuracy constraint(10d). Our goal is to select a subset of clients C (1<C≤N) for complete training in each communication round to maximize the total accuracy gain $\overset{T}{\underset{t=1}{\sum }}\Delta A_{t}$ while minimizing the total latency $\overset{T}{\underset{t=1}{\sum }}H_{t}$ and total communication cost $\overset{T}{\underset{t=1}{\sum }}B_{t}$ of the entire FL training. Thus, the proposed optimization is NP-hard.

To solve problem (10), our FedDdrl algorithm adopted a double DRL framework for our optimization problem. Specifically, we formulated the DRL policy network for both tasks using a multi-agent reinforcement learning (MARL) approach. In particular, VDN has proven itself in the recent literature [33] to be a promising candidate for MARL problems. A VDN network consists of N agents, in which each agent n∈N uses a deep neural network (DNN) parametrized with θ to implement the Q-function $Q_{n}^{\theta }\left(s,a\right)=E[R_{t}|s=s_{n}^{t},a=a_{n}^{t}]$. At each timestep t, each agent n observes its states $s_{n}^{t}$ and selects the optimal action $a_{n}^{t}$ with the maximum Q-value. Let $s_{t}=\left\{s_{n}^{t}\right\}$ and $a_{t}=\left\{a_{n}^{t}\right\}$ represent the states and actions collected from all agents n∈N at timestep t, respectively. The joint Q-function $Q_{tot}\left(\cdot \right)$ for the multi-agent VDN system can be represented by the elementwise summation of all the individual Q-functions, where $Q_{tot}\left(s_{t},\;a_{t}\right)=\sum_{n}Q_{n}^{\theta }\left(s_{n}^{t},a_{n}^{t}\right)$. In FedDdrl, we set each agent in both VDN as a simple two-layer multi-layer perceptron (MLP), which is cheap to implement. All MLPs in each VDN share their weights to prevent the lazy agent problem [33].

As illustrated in Figure 6, the first VDN network takes the client states $s_{t}$ (which will be detailed later) to obtain the optimal client termination matrix $\phi_{t}$. The second VDN network takes the same client states $s_{t}$ to obtain the optimal local training epoch per client $E_{t}$.

#### 4.1.1. Early Client Termination

Inspired by the work of FedMarl [14], the first VDN network was employed to learn the optimal policy for early client termination matrix $\phi_{t}$ at round t. We reformulated the problem as an MDP with the following state, action and reward to train a VDN network with N=10 agents.
**State**$s_{t}$
State $s_{t}=\left\{s_{n}^{t}\right\}$ consisted of the client states for each VDN agent. Each agent n consisted of six components: (i) the probing loss $L_{n}^{t}$, (ii) probing training latencies $H_{t,n}^{p}$, (iii) model uploading latencies $H_{t,n}^{u}$, (iv) communication cost from client to server $B_{n}^{t}$, (v) local training dataset size $\left|D_{n}\right|$ and (vi) current communication round index t. The state vector for agent c can be written as Equation (11):(11)$$s_{n}^{t}=\left[L_{n}^{t},H_{t,n}^{p},H_{t,n}^{u},B_{n}^{t},\left|D_{n}\right|,t\;\right]$$It is noteworthy that since each agent in the VDN only has access to its own local observation instead of the full observed environment, the policy has to incorporate past agent observations from history [33]. Thus, the historical values of probing latencies $H_{t,n}^{p}=\left[H_{t-\Delta T_{p},n}^{p},\;\ldots ,\;H_{t,n}^{p}\right]$ and model uploading latencies $H_{t,n}^{u}=\left[H_{t-\Delta T_{u}-1,n}^{u},\;\ldots ,\;H_{t-1,n}^{u}\right]$ were included in the state vector to mitigate the limitation of local observation. Note that $\Delta T_{p}$ and $\Delta T_{u}$ are the sizes of the historical information of probing latencies and model uploading latencies, respectively.**Action** $\phi_{t}$: Action $\phi_{t}=\left\{\phi_{n}^{t}\right\}$ comprised the client termination decision for each VDN agent. The action space for client termination was $\phi_{n}^{t}=\left\{0,\;1\right\}$, where 0 indicates the termination of the client and 1 indicates the client is selected for complete training.**Reward** $r_{t}^{1}$: A vanilla reward for VDN 1, denoted as $r_{t}^{1}$, can be adopted from the FL optimization problem as described in Equation (12):(12)$$r_{t}^{1}=w_{1}U\left(\Delta A_{t}\right)-\left(w_{2}B_{t}+w_{3}H_{t}\right)$$where the system is rewarded with accuracy improvement $\Delta A_{t}$ and penalties for $B_{t}$ and $H_{t}$. However, Equation (12) has one obvious limitation. When $\Delta A_{t}\to 0$, the $\underset{\Delta A_{t}\to 0}{\lim}w_{1}U\left(\Delta A_{t}\right)=0$, regardless of the magnitude of $w_{1}$. If $w_{1}U\left(\Delta A_{t}\right)\to 0$, the reward $r_{t}^{1}\approx -\left(\bar{w_{2}B_{t}+w_{3}H_{t}}\right)$. This causes the optimization problem to diverge from improving accuracy with the constraint of $B_{t}$ and $H_{t}$ to merely the reduction of $B_{t}$ and $H_{t}$. To show the severeness of this problem, we trained the VDN agents using the $r_{t}^{1}$ as defined by Equation (12). Let $E\left[w_{1}U\left(\Delta A_{t}\right)\right]$ and $E\left[w_{2}B_{t}+w_{3}H_{t}\right]$ denote the expected values of accuracy improvement $\Delta A_{t}$ and penalties ($w_{2}B_{t}+w_{3}H_{t})$, respectively. For MNIST dataset, the expected values of both components for the last R=5 communication rounds can be computed in Equations (13) and (14):(13)$$\begin{array}{l} {\left.E\left[w_{1}U\left(\Delta A_{t}\right)\right]\right|}_{t=\left\{T-R,\;T-R-1,\;\ldots T\right\}}\\ =\frac{1}{5}\overset{T}{\underset{t=T-5}{\sum }}w_{1}U\left(\Delta A_{t}\right)\\ =0.0273 \end{array}$$
(14)$$\begin{array}{l} {\left.E\left[w_{2}B_{t}+w_{3}H_{t}\right]\right|}_{t=\left\{T-R,\;T-R-1,\;\ldots T\right\}}\\ =\frac{1}{5}\overset{T}{\underset{t=T-5}{\sum }}\left(w_{2}B_{t}+w_{3}H_{t}\right)\\ =0.279 \end{array}$$It is observed that $E\left[w_{1}U\left(\Delta A_{t}\right)\right]\ll E\left[w_{2}B_{t}+w_{3}H_{t}\right]$ for the last five communication rounds. This is because as training approach the end, the accuracy improvement is often smaller compared to the earlier stage. Consequently, the VDN agents start to terminate more clients from complete training, giving way to the reduction of $B_{t}$ and $H_{t}$. To make sure the agents are motivated to learn even when $\Delta A_{t}\to 0$, we can introduce a bias term b to $r_{t}^{1}$. Let $b=\frac{3}{10}\;E\left[w_{2}B_{t}+w_{3}H_{t}\right]$. Hence, the reward function $r_{t}^{1}$ can be reformulated as shown in Equation (15):(15)$$r_{t}^{1}=\left\{\begin{array}{c} r_{t}+b,\;\;\;\;\;\;\Delta A_{t}>0\\ \;\;r_{t},\;\;\;\;\;\;\;\;\;\;\Delta A_{t}\leq 0 \end{array}\right.\;,\;r_{t}=w_{1}U\left(\Delta A_{t}\right)-\left(w_{2}B_{t}+w_{3}H_{t}\right)$$Note that we only added the bias term b to the reward $r_{t}$ when $\Delta A_{t}>0$ since it is intended to encourage accuracy improvement. We did not subtract the bias term b from the reward $r_{t}$ when $\Delta A_{t}\leq 0$ since the penalty terms are sufficient to penalize the inferior actions.

#### 4.1.2. Local Epoch Adjustments

The second DRL network was employed to learn the optimal policy for local epoch adjustments $E_{t}$ at round t. A VDN algorithm with N=10 agents can be formulated by defining the state, action and reward as follows:**State**$s_{t}$: The second VDN shared the same state in Equation (11) since both VDNs required the same local observation for decision making.**Action** $E_{t}$: Action $E_{t}=\left\{E_{n}^{t}\right\}$ comprises the local epoch counts for each VDN agent. The action space is $E_{n}^{t}=\left\{3,5,\;7\right\}$. This action aims to exploit client devices with stronger computation power for more training epochs and vice versa.**Reward** $r_{t}^{2}$: We adopted Equation (15) as the starting point for the reward function for VDN 2. However, the communication cost $B_{t}$ was not part of the optimizing objectives of VDN 2 since local epoch adjustment is only bounded by the $H_{t}$ constraint. Hence, the reward function $r_{t}^{2}$ for this VDN networks can ignore the $B_{t}$ penalty. As such, the $r_{t}^{2}$ can be defined in Equation (16):(16)$$r_{t}^{2}=\left\{\begin{array}{c} r_{t}+b,\;\;\Delta A_{t}>0\\ \;\;r_{t},\;\;\;\;\;\;\Delta A_{t}\leq 0 \end{array}\right.\;,\;where\;r_{t}=w_{1}U\left(\Delta A_{t}\right)-w_{3}H_{t}$$where we used the same bias term b from (15) for the simplicity’s sake.

As the training converges, VDN 1 will deliver the optimal client selection, and VDN 2 will impart the optimal local epoch number for each client. The overall algorithm for solving the problem in Equation (10a–e) is summarized in Algorithm 2.
**Algorithm 2** FedDdrl Algorithm1:**Input:**Initialize VDN 1 $Q_{tot}^{1}$ and its target network $Q_{tot}^{1}{}^{’}$ for client selection $\phi_{t}$ policy 

Initialize VDN 2 $Q_{tot}^{2}$ and its target network $Q_{tot}^{2}{}^{’}$ for local epoch adjustment $E_{t}$ policy2:**Output:**Trained $Q_{tot}^{1}$ and $Q_{tot}^{2}$ networks3:Set ε=1.0
4:**for** Episode $n_{ep}=1,\;2,\;\ldots ,\;N_{ep}$ **do**5:
Reset the FL environment6:
Initialize a global model $w_{0}$
7:
**for** communication round t=1,2,…,T **do**8:

Randomly select N clients from all K clients9:

Broadcast the global model $w_{t}$ to each selected client10:

**for** each client n∈N in parallel **do**11:


$w_{t}^{n}\leftarrow w_{t}$; Copy the global model as each client model12:


Update the client model $w_{t}^{n}$ using the local training dataset $D_{n}$
13:


Upload client states $s_{n}^{t}$ to the FL server14:

**end for**15:

Each agent n in VDN 1 selects the optimal action $\phi_{t,n}^{*}$= argmax $Q_{n}^{1}\left(s_{n}^{t},\phi_{n}^{t}\right)$ with a $\left(1-\varepsilon \right)$ × 100% probability, else randomly output actions16:

Each agent n in VDN 2 selects the optimal action $E_{t,n}^{*}$= argmax $Q_{n}^{2}\left(s_{n}^{t},E_{n}^{t}\right)$ with a $\left(1-\varepsilon \right)$ × 100% probability, else randomly output actions17:

Send action $\phi_{t,n}^{*}$ and $E_{t,n}^{*}$ to each client n∈N
18:

**for** each client n∈N in parallel **do**19:


**if** $\phi_{t,n}^{*}=1$:20:



Continue updating $w_{t}^{n}$ using $D_{n}$ until $E_{t,n}^{*}$ is reached21:



Return updated $w_{t}^{n}$
22:


**end if**23:

**end for**24:

Aggregate global model $w_{t+1}\leftarrow \sum_{i=1}^{N}\frac{\left|D_{i}\right|}{\sum_{i=1}^{N}\left|D_{i}\right|}w_{t}^{n}$ where $i\in \{n|\phi_{t,n}^{*}=1\}$
25:

Reward $r_{t}^{1}$ and $r_{t}^{2}$ are given to $Q_{tot}^{1}$ and $Q_{tot}^{2}$ based on $\Delta A_{t},B_{t},H_{t}$
26:

$s_{t}=\left\{s_{n}^{t}\right\}$, $\phi_{t}=\left\{\phi_{n}^{t}\right\}$, $E_{t}=\left\{E_{n}^{t}\right\}$27:

Store transitions 1 $\left[s_{t},\phi_{t},r_{t}^{1}\right]$ for $Q_{tot}^{1}$ into memory buffer 128:

Store transitions 2 $\left[s_{t},E_{t},r_{t}^{2}\right]$ for $Q_{tot}^{2}$ into memory buffer 229:

Sample mini-batches with size $n_{b}$ from memory buffer to train $Q_{tot}^{1}$,$\;Q_{tot}^{2}$ and $Q_{tot}^{1}{\;}^{’},\;Q_{tot}^{2}{\;}^{’}$
30:

Decay ε gradually from 1.0 to 0.131:
**end for**32:**end for**

### 4.2. Balanced-MixUp to Mitigate Weight Divergence

In this study, we focused on the non-IID label shift, where the client dataset is heavily skewed to one of the label classes. The huge EMD between the client data distribution $p^{k}$ and the global distribution p will contribute to weight divergence, deteriorating the training efficiency of FL. Thus, the highly imbalanced client dataset has to be handled wisely. MixUp [34] is a simple yet effective data augmentation technique that could shed some light on this problem.

MixUp extends the training data distribution by linearly interpolating between existing data points, filling the underpopulated areas in the data space. It generates synthetic training data $\left(\hat{x},\;\hat{y}\right)$ by simply taking the weighted combination of two random data pairs, $\left(x_{i},\;y_{i}\right)$ and $\left(x_{j},\;y_{j}\right)$, as shown in Equations (17) and (18):(17)$$\hat{x}=\lambda x_{i}+\left(1-\lambda \right)x_{j}$$
(18)$$\hat{y}=\lambda y_{i}+\left(1-\lambda \right)y_{j}$$
where λ ~ Beta(α, α), with α > 0. Despite its simplicity, MixUp has been proven to improve model calibration and better generalization [35]. Thus, its application has expanded from image and speech classification tasks [34] to other domains, including image segmentation [36] and natural language processing [37,38]. However, a vanilla MixUp works poorly in highly imbalanced datasets [39]. In highly imbalanced datasets, MixUp would end up sampling the data pairs $\left(x_{i},\;y_{i}\right)$ and $\left(x_{j},\;y_{j}\right)$ from the same class for most of the time since the sampling is done randomly.

Some recent studies focused on solving the data imbalance problem for MixUp. In Remix by [39], $x_{i}$ and $x_{j}$ are mixed in the same fashion as MixUp, but $y_{i}$ and $y_{j}$ are mixed such that the minority class is assigned a higher weight. This method pushes the decision boundaries away from the minority class, balancing the generalization error between the majority and minority classes. Balanced-MixUp is another variation of MixUp, where MixUp is combined with a data-resampling technique to achieve a balanced class distribution [22]. Specifically, balanced-MixUp combines instance-based sampling and class-based sampling for the majority and minority classes, respectively. This ensures that each data pair $\left(x_{i},\;y_{i}\right)$ and $\left(x_{j},\;y_{j}\right)$ always consists of instances from both the majority and minority classes.

We adopted balanced-MixUp as the augmentation into the formulation of our solution to address the class imbalanced problem. To the best of our knowledge, we are the first to integrate balanced-MixUp into FL for weight divergence mitigation. Let $\left(x_{M},\;y_{M}\right)$ and $\left(x_{m},\;y_{m}\right)$ denote the instance pair sampled from the majority and minority classes, respectively. Balanced-MixUp can be expressed as shown in Equations (19) and (20):(19)$$\hat{x}=\lambda x_{M}+\left(1-\lambda \right)x_{m}$$
(20)$$\hat{y}=\lambda y_{M}+\left(1-\lambda \right)y_{m}$$

Balanced-MixUp guarantees that each data pair mixing consists of instances from both the majority and minority class. Unlike the original balanced-MixUp where λ ~ Beta(1, α), we adopted λ ~ Beta(α, α) and found it to work better in our study. The best α value may be different depending on the datasets, which will be detailed in the results section.

## 5. Simulation Results

This study adopted TensorFlow as the deep learning platform. We adopted three datasets for FL benchmarking: MNIST, CIFAR-10 and CrisisIBD [40]. First, MNIST is a relatively simple task under most non-IID settings. It is mainly used to prove that a novel FL algorithm is at least working. In contrast, CIFAR-10 is a challenging dataset in non-IID settings, which is strongly encouraged to be included in FL benchmarking experiments [32]. On the other hand, the CrisisIBD dataset is the benchmark dataset for various real-world disaster-related image classification tasks. In this study, we adopted the disaster classification dataset from the dataset (hereinafter referred to as CrisisIBD) as one of our benchmark datasets. We adopted balanced-MixUp to augment all three datasets. We found the best α value used by balanced-MixUp was 0.05, 0.4 and 0.2 for MNIST, CIFAR-10 and CrisisIBD, respectively. We used all three datasets to train a lightweight MobileNetV2 [41], which aligned with our goal of developing the FL framework for low-powered IoT devices. All clients adopted $B_{n}^{t}=1\forall n,t$, similar to the setting in [14].

In this study, we focused on the non-IID label shift as demonstrated in [14,21]. Similarly, we divided each dataset into K clients. For each client, a fraction σ=0.8 of the local training dataset was sampled from one random label (which is the majority label), while the rest of the training data were sampled uniformly from the remaining labels (which are the minority labels). We compared our proposed method with FedAvg, FedProx and FedMarl. The first two algorithms were shown to be robust baselines in non-IID label shift [32] and are prebuilt in many existing FL frameworks, including Tensorflow Federated [42] and Intel OpenFL [8]. On the other hand, FedMarl is one of the state-of-the-art FL algorithms [14]. Our FedDdrl aims to outperform all three of the algorithms. The hyperparameters are listed in Table 3.

Due to limited resources, we only had two hardware devices: (i) an Intel NUC with an Intel core i7-10710U processor with 4.70 GHz and (ii) a workstation equipped with an Intel core i7-10875H processor with 2.30 GHz and NVIDIA RTX 2070 SUPER. In total, this yielded two TensorFlow CPU operators and one TensorFlow GPU operator. However, this was far from enough to simulate a heterogeneous FL environment with K=100 clients if each operator only represents one client. Thus, we carefully devised our experimental setup, as shown in Figure 7.

To simulate an FL environment with K=100 clients (IoT devices) and C=10 selected clients (before early termination), we set up the experiment as shown below:We created K=100 client configurations, each consisting of the (i) client’s computing latency per data, (ii) model upload latency and (iii) local dataset identity (ID) number. To closely simulate the heterogeneity of resources in an IoT network as in [14], the computing latency per data in each client configuration can be any of $\left\{0.25,\;0.50,\;0.75\right\}$ seconds, while the model upload latency can be any of $\left\{1.00,\;1.25,\;1.75,\;2.00\right\}$ seconds.CPU 1, CPU 2 and GPU simulated three, three and four clients, respectively. The simulated clients represent the C=10 randomly selected clients from the total K=100 clients in each communication round t.In each communication round t, 10 client configurations were randomly sampled out from the configuration pools. The 10 simulated clients (in CPU 1, CPU 2 and GPU) were configured according to the selected client configuration. This entire process (3) is equivalent to the FL process of randomly selected 10 clients with unique local datasets and resources.After step (3), each simulated client proceeded with its training. If the FL algorithm was FedAvg or FedProx, all 10 simulated clients underwent complete training of $E_{t}=5$ local epochs. On the contrary, if the FL algorithm was FedMarl or FedDdrl, only the simulated clients that were not terminated by the FedMarl/FedDdrl completed their local training based on $E_{t,n}^{*}$= argmax $Q_{n}^{2}\left(s_{n}^{t},E_{n}^{t}\right)$ by VDN 2.

### 5.1. Results and Ablation Study

We compared the performance of FedDdrl with other baselines in all three objectives of the optimization problem, which are the (i) model accuracy, (ii) training latency and (iii) communication efficiency.

#### 5.1.1. Model Accuracy

Table 4 shows the model accuracy trained using each FL setting after T=15 communication rounds. We also conducted an ablation study showing how FedDdrl improves beyond FedMarl.

Setting A in Table 4 was our implementation of FedMarl with the original hyperparameters ($w_{1}$ = 1.0, $w_{2}$ = 0.1, $w_{3}$ = 0.2). In Setting B, we proved that FedMarl performance could be improved using our hyperparameters ($w_{1}$ = 2.9, $w_{2}$ = 0.1, $w_{3}$ = 0.2). However, we found that its accuracy was still far behind both FedAvg and FedProx. This is because MobileNetV2 is a lightweight model which is easy to overfit [41]. In FedAvg and FedProx, the total number of clients selected for training is always C=10 in each communication round t. During aggregation, there is a sufficient amount of client models overfitted for different classes, which, when aggregated, can generate a regularization effect, thus mitigating the weight divergence caused by overfitting. This is not the case for FedMarl, which does not have a fixed C for each round. We argue that since the original FedMarl was not tested on MobileNetV2, it was able to perform better than FedAvg and FedProx. Applying balanced-MixUp could significantly mitigate this problem, as shown in Setting C. The reasoning on how balanced-MixUp helps weight divergence mitigation is detailed in Section 5.5.

Setting D was our FedDdrl, where we added another VDN for local epoch adjustment. FedDdrl allows client devices to train for more epochs when required and vice versa. This allows FedDdrl to converge faster than FedAvg and FedProx for most cases, even when it is not utilizing all clients at each communication round. FedDdrl outperformed other FL algorithms in both the challenging CIFAR-10 and CrisisIBD datasets, and it was the second-best for MNIST. We suspect FedDdrl is slightly overengineered for an easy task like MNIST. Nevertheless, it was still very robust considering that real-world data is often not as simple as MNIST and is instead more challenging like the CIFAR-10 and CrisisIBD datasets.

#### 5.1.2. Training Latency

Figure 8 shows the normalized training latency for each FL algorithm (with balanced-MixUp) on all three datasets. Our FedDdrl outperformed all three other algorithms in all datasets. This is promising since FedDdrl allows dynamic local adjustment. The FedDdrl will sometimes increase the local epoch from five to seven. However, the extra training latency is balanced when FedDdrl decreases the local epoch from five to three, especially in the early communication round when the MobileNetV2 is still learning lower-level features. FedMarl followed closely behind FedDdrl. This is mainly because both FedDdrl and FedMarl can terminate clients with longer probing latencies. On the other hand, FedAvg had a moderate performance in terms of training latency. It was not as fast as FedMarl and FedDdrl, but it was still significantly faster than FedProx. As expected, FedProx had the longest training latency compared to other FL algorithms, which is aligned with the observation by [32]. This is mainly due to the extra computing cost required to compute the L2 distance between the client and the global model. Hence, applying FedProx in low-powered devices (i.e., IoT devices) with limited computation power is not feasible.

#### 5.1.3. Communication Efficiency

Figure 9 shows the normalized communication costs of all FL algorithms in the three datasets. FedDdrl and FedMarl were significantly more efficient than FedAvg and FedProx in total communication costs. There was no clear winner between FedDdrl and FedMarl regarding communication efficiency. However, FedDdrl outperformed FedMarl in CIFAR-10 and CrisisIBD datasets, which are significantly harder tasks compared to MNIST. Thus, we argue that FedDdrl is the best algorithm. On the other hand, FedAvg and FedProx have a fixed number of clients selected in each round. Since we assume $B_{n}^{t}=1\forall n,t$, both algorithms have the same total communication costs.

### 5.2. Strategy Learned by FedDdrl

In this section, we analyze the strategies learned by the FedDdrl algorithm to fully utilized its computing resources while reducing communication costs. Figure 10 and Figure 11 show the early client termination strategy learned by FedDdrl. Blue dots indicate that the client was chosen for complete training, while red dots indicate early client termination.

First, FedDdrl generally picked lesser clients for complete training in the early phase of FL training. From Figure 10 and Figure 11, it is noticed that only three clients were selected for complete training in the first two communication rounds $t=\left\{1,2\right\}$. This is because DNNs usually learn the low-complexity features before learning the higher-complexity features. The former is more robust to noises [43] and can be learned with fewer data [14]. This allows FedDdrl to reduce communication costs by terminating most clients from training in the early phase, where the MobileNetV2 is still learning low-level features. Starting from round t=3 to t=6, the number of clients that underwent complete training increased from 6 to 10. This indicates that MobileNetV2 was beginning to learn higher-level features that require more training data. For the remaining rounds, the number of selected clients was roughly five. Second, FedDdrl preferred clients with a lower probing loss for complete training, which is aligned with the findings in FedMarl [14]. Third, FedDdrl tended to pick clients with shorter probing latency for complete training to reduce the total latency of FL training.

As mentioned earlier, conventional FL training sets the same local epoch $E_{t}^{n}$ for all clients, disregarding their computing resources. Hence, one of the contributions of FedDdrl is to learn the optimal strategy to adjust the local epoch count for each client dynamically. In Figure 12, we plotted the local epoch count corresponding to each selected client from the scenario in Figure 11. Bigger dots indicate that a higher local epoch count was assigned for the corresponding clients. It was found that FedDdrl tended to set a lower epoch count (smaller dots) for clients with higher probing latency. This method could reduce the total training latency since clients with limited computing power did not have to participate in long training epochs. On the other hand, clients with lower probing latency tended to have a higher epoch count. This strategy can fully utilize the computing resource of clients with stronger computing power since they can continue training while waiting for other clients to finish. However, this was not always the case, as shown in Figure 13. On certain occasions, FedDdrl set a high epoch count for clients with long probing latency if the data in these clients were crucial for FL convergence. In any case, FedDdrl was superior to FedMarl, where the FedMarl could either select or terminate a client without the third option of selecting the clients and dynamically tuning the local epoch.

### 5.3. How FedDdrl Optimizes the Three Objectives Simultaneously

The objectives of FedDdrl are to (i) maximize the global model’s accuracy while minimizing the (ii) FL system’s training latency and (iii) communication cost. First, VDN 1 will perform early client termination to terminate clients who are not essential for training. By doing so, we can reduce the total communication cost. Additionally, VDN 1 prefers clients with lower probing latency (which also translates to lower training latency). Thus, VDN 1 plays a huge role in reducing both communication costs and training latency. Second, VDN 2 will dynamically adjust the local epoch count. Clients with limited computing power only have to train with a lesser epoch, so they can finish training earlier. Meanwhile, VDN 2 assigns a higher epoch count to clients with stronger computing power, so they can continue training while waiting for the slower clients. Hence, VDN 2 reduces the training latency and fully utilizes clients with stronger computing power.

Lastly, the global model’s accuracy must be retained. When VDN 1 performs client termination, it is essentially reducing C. Intuitively, reducing C seems counter-productive since it reduces the total number of clients participating in training for each communication round t. Fewer clients translate to fewer training data. However, [44] showed that in a non-IID setup, the convergence rate of FedAvg had a weak dependence on C. This makes sense, as some clients may have local datasets with a huge EMD distance from the global distribution. Training the FL model using these clients may hinder the convergence rate. Additionally, [14] showed that using DRL in client selection (or early client termination) can positively affect the convergence rate. This is because selecting useful clients (with useful data) can improve the quality of the overall FL data, which is more crucial than increasing the quantity of data.

In short, FedDdrl can reduce communication cost and training latency without sacrificing model accuracy via early client termination due to the weak correlation between convergence rate and C.

### 5.4. Computational Complexity Analysis

FedDdrl is composed of a finite number of MLPs. In MLP, let L, $n_{0}$ and $n_{i}$ denote the layer numbers, the size of the input layer (which corresponds to the client state’s size) and the number of neurons in i-th layer, respectively. During training mode, the computational complexity for an MLP to update its weight in each step can be expressed as $O(N_{b}(n_{0}n_{1}+\sum_{i=1}^{L-1}n_{i}n_{i+1}))$ [45]. In total, it takes $N_{ep}$ × T steps for the FedDdrl algorithm to finish training. Hence, the total training computational complexity of FedDdrl is $O(N_{ep}TN_{b}(n_{0}n_{1}+\sum_{i=1}^{L-1}n_{i}n_{i+1}))$. The high computation complexity of the MLP can be performed offline using a powerful device (i.e., the FL server). In the online deployment mode, the computational complexity in each step is dramatically reduced to $O(n_{0}n_{1}+\sum_{i=1}^{L-1}n_{i}n_{i+1})$. This is done by cutting off the training procedure, which requires feedforward and backpropagation of $N_{b}$ data points. Thus, the computational complexity is retained at a favorable level.

### 5.5. Why Balanced-MixUp Helps in Federated Learning

Without loss of generality, we explored how balanced-MixUp mitigates weight divergence in FL assuming the amount of training data for each class is uniform in the global population. Under this setting, we can express the global distribution $p\left(y=i\right)$ for all labels $i=\left\{1,\;2,\;3,\;\ldots ,\;n_{c}\right\}$ as shown in Equation (21):(21)$$p\left(y=i\right)=\frac{1}{n_{c}}$$

In this study, a fraction σ of the local training dataset is sampled from one random label, while the remaining 1−σ fraction is sampled uniformly from the remaining labels. Following this assumption, let i=1 be the majority class in each client and $i=\left\{2,\;3,\;\ldots ,\;n_{c}\right\}$ be the minority classes (whichever i can be the majority class since the ordering does not affect the approximation of client distribution). Without balanced-MixUp, we can express the client dataset distribution $p^{k}\left(y=i\right)$ as Equation (22):(22)$$p^{k}\left(y=i\right)=\left\{\begin{array}{l} \;\;\;\;\sigma ,\;\;\;\;\;\;i=1\\ \frac{1-\sigma }{n_{c}-1},\;\;\;\;i=\left\{2,\;3,\;\ldots ,\;n_{c}\;\right\} \end{array}\right.$$

Based on Equations (21) and (22), the EMD between local and global distribution for FedAvg without balanced-MixUp, denoted as ${\mathrm{EMD}}_{ori}$, can be written as Equation (23):(23)$$\begin{array}{l} {\mathrm{EMD}}_{ori}\\ =\sum_{i=1}^{n_{c}}∥p^{k}\left(y=i\right)-p\left(y=i\right)∥\\ =∥\sigma -\frac{1}{n_{c}}∥+\left(n_{c}-1\right)∥\frac{1-\sigma }{n_{c}-1}-\frac{1}{n_{c}}∥ \end{array}$$

On the other hand, the client dataset distribution with balanced-MixUp $p_{MixUp}^{k}\left(y=i\right)$ can be expressed as shown in Equation (24):(24)$$p_{MixUp}^{k}\left(y=i\right)=\left\{\begin{array}{l} \;\;\;\;E\left(\lambda \right),\;\;i=1\\ \frac{1-E\left(\lambda \right)}{n_{c}-1},\;\;\;\;i=\left\{2,\;3,\;\ldots ,\;n_{c}\;\right\} \end{array}\right.$$
where $E\left(\lambda \right)$ is the expected value of λ ~ Beta(α, β). $E\left(\lambda \right)$ can be written as Equation (25):(25)$$E\left(\lambda \right)=\frac{\alpha }{\alpha +\beta }$$

Based on Equations (21) and (24), the EMD between FedAvg with balanced-MixUp denoted as ${\mathrm{EMD}}_{\mathrm{MixUp}}$ can be written as Equation (26):(26)$$\begin{array}{l} {\mathrm{EMD}}_{\mathrm{MixUp}}\;\\ =∥\sum_{i=1}^{n_{c}}p_{MixUp}^{k}\left(y=i\right)-p\left(y=i\right)∥\\ =∥E\left(\lambda \right)-\frac{1}{n_{c}}∥+\left(n_{c}-1\right)∥\frac{1-E\left(\lambda \right)}{n_{c}-1}-\frac{1}{n_{c}}∥ \end{array}$$

Take our experiments using CIFAR-10 as example, where σ=0.8, $n_{c}=10$, λ ~ Beta(0.4, 0.4) and $E\left(\lambda \right)=0.5$. Based on Equations (23) and (26), the ${\mathrm{EMD}}_{ori}$ and ${\mathrm{EMD}}_{\mathrm{MixUp}}$ can be computed as 0.777 and 0.444, respectively. This shows that balanced-MixUp could significantly reduce the weight divergence caused by the EMD between $p^{k}$ and p.

The above proposition is aligned with our experiments. Table 5 shows the performance of FedAvg with and without balanced-MixUp in different C. Balanced-MixUp provided a drastic accuracy boost to FedAvg in all datasets. The accuracy improvement was as high as 17.0% for the CrisisIBD dataset when C=10. This shows that balanced-MixUp can effectively mitigate the weight divergence caused by the EMD, especially when a low number of clients participate in a communication round t. Note that for the MNIST dataset (C=10), the accuracy of FedAvg with balanced-MixUp was slightly poorer than its counterpart, lagging behind by merely 1.4%. This is reasonable, as MNIST is considered a simple task [32] in which FedAvg could perform similarly in certain non-IID settings. Another notable observation is that FedAvg with balanced-MixUp significantly outperformed its counterpart in both the challenging CIFAR-10 and CrisisIBD datasets. The observation is consistent in both C=5 and C=10 experiments. This is encouraging because it proves that balanced-MixUp is useful in mitigating non-IID label shifts, especially for algorithms like FedMarl and FedDdrl which do not have a fixed value of C.

## 6. Conclusions

In this paper, we proposed a DDRL-based FL framework (FedDdrl) for adaptive early client termination and local epoch adjustment. FedDdrl can terminate clients with high probing latency to reduce total training latency and communication costs, and it can automatically adjust the local epoch to fully utilize clients’ computing resources. We also showed that balanced-MixUp is a useful augmentation technique to mitigate the impact of weight divergence arising from non-IID label shifts in FL. The simulation results on MNIST, CIFAR-10 and CrisisIBD confirmed that FedDdrl outperformed the comparison schemes in terms of the model’s accuracy, training latency and communication costs of FL under extreme non-IID settings. As a future work, we would explore the performance of FedDdrl on other types of non-IID settings, such as feature distribution skew and quantity skew.

## Figures and Tables

**Figure 1 sensors-23-02494-f001:**
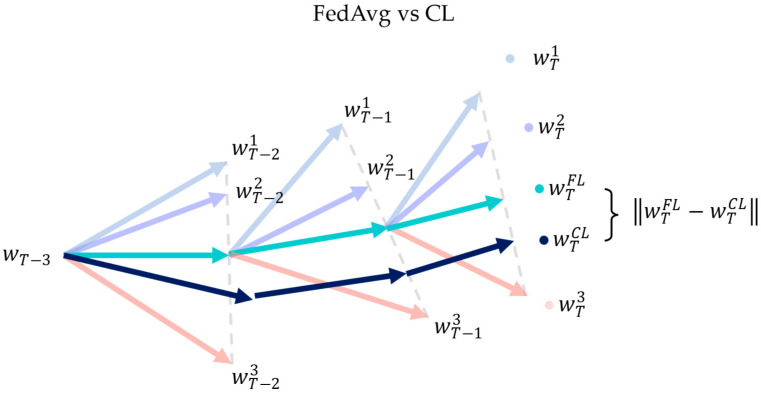
Weight divergence between $w_{t}^{FL}$ and $w_{t}^{CL}$ is inevitable even if both models have the same initialization weights.

**Figure 2 sensors-23-02494-f002:**
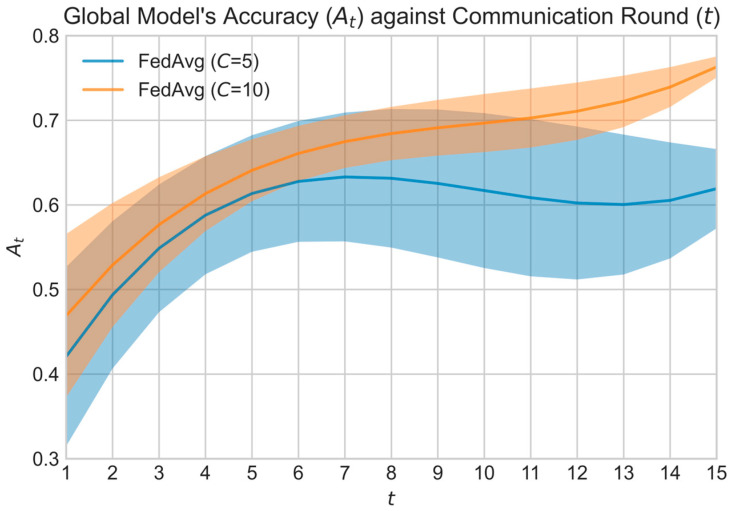
The global model’s accuracy curve for C=5 and C=10.

**Figure 3 sensors-23-02494-f003:**
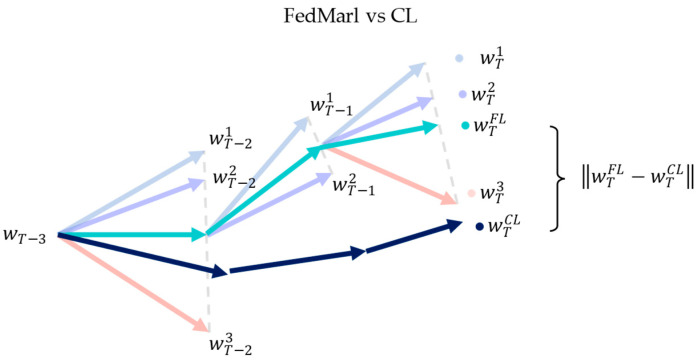
Ineffective client dropping by FedMarl will lead to large weight divergence. Client 3 (denoted by peach) is dropped from training at communication round T−2. This causes the aggregated FL weights $w_{t}^{FL}$ (denoted by green) to converge toward Client 1 and 2 while diverging away from the $w_{t}^{CL}$.

**Figure 4 sensors-23-02494-f004:**
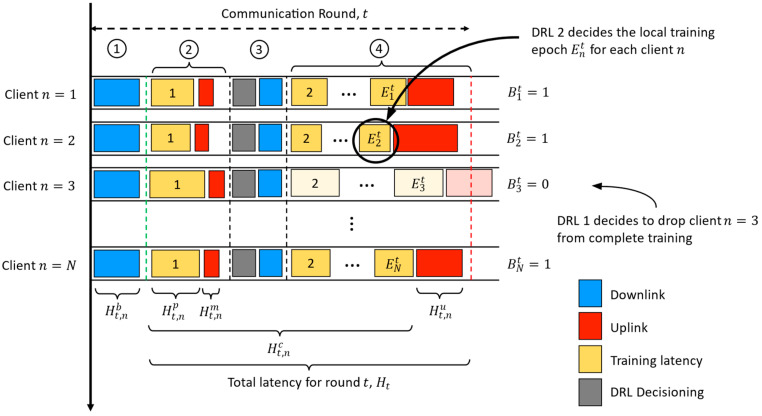
The proposed FL protocol’s system model consists of four phases in each communication round: (1) model broadcasting, (2) probing training, (3) early client termination and (4) completion of training.

**Figure 5 sensors-23-02494-f005:**
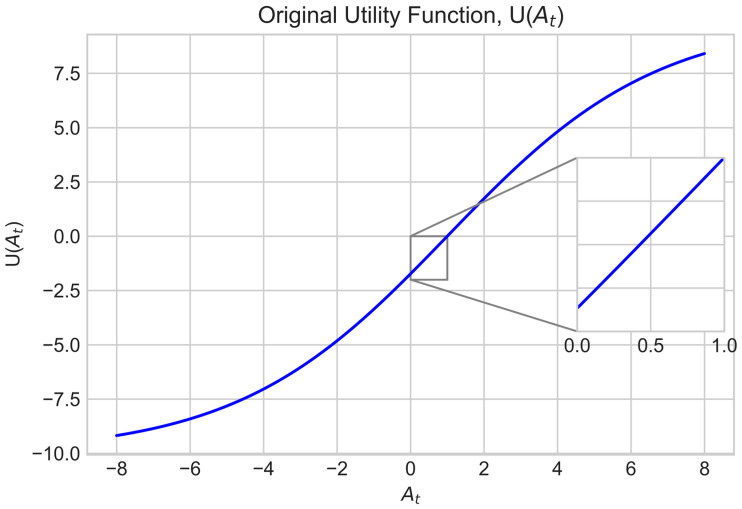
The original utility function $U\left(A_{t}\right)$ can be approximated as a straight line when $A_{t}$ falls in the range [0, 1].

**Figure 6 sensors-23-02494-f006:**
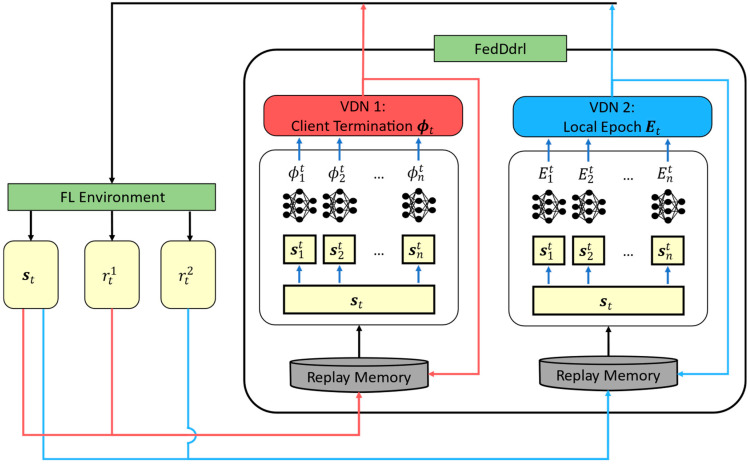
Structure of the proposed FedDdrl algorithm.

**Figure 7 sensors-23-02494-f007:**
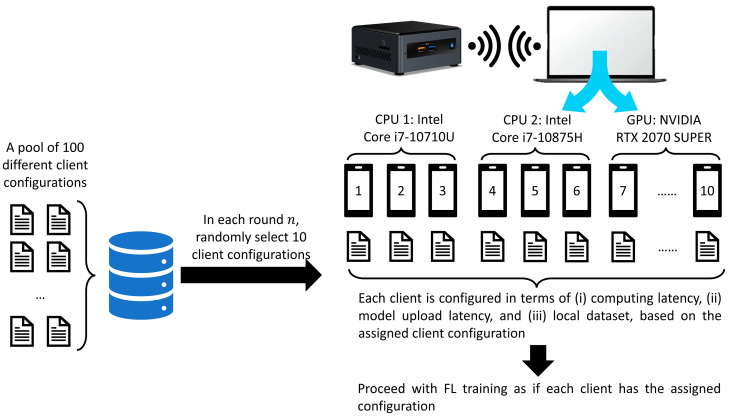
Experiment setup.

**Figure 8 sensors-23-02494-f008:**
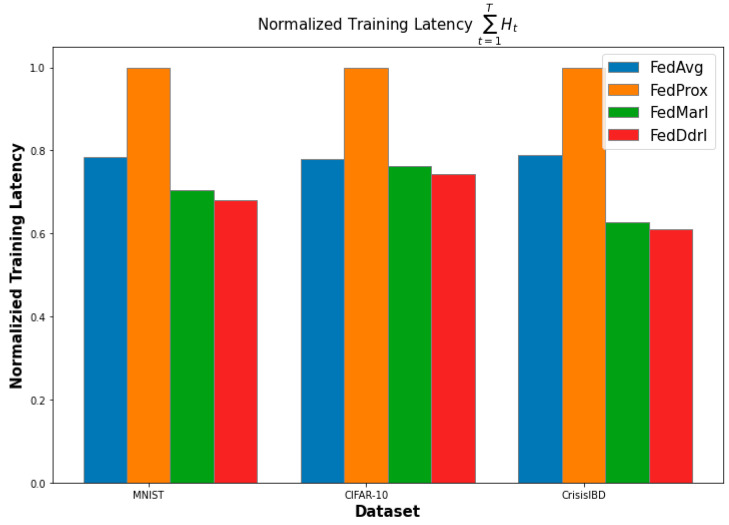
Normalized training latency of each FL algorithm.

**Figure 9 sensors-23-02494-f009:**
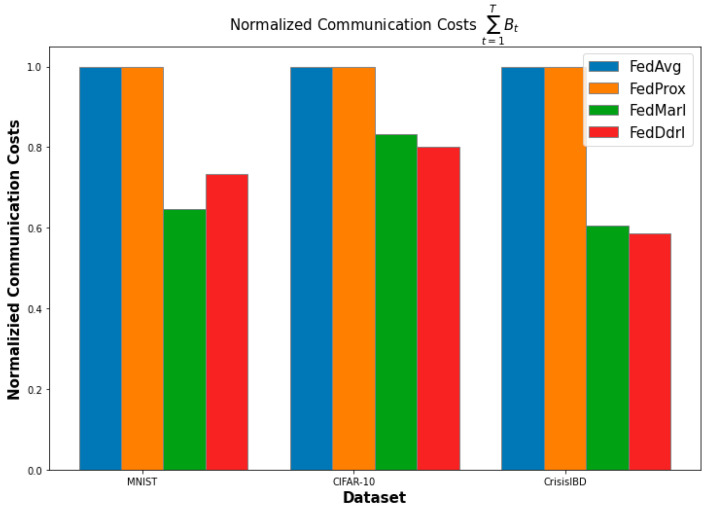
Normalized communication cost of each FL algorithm.

**Figure 10 sensors-23-02494-f010:**
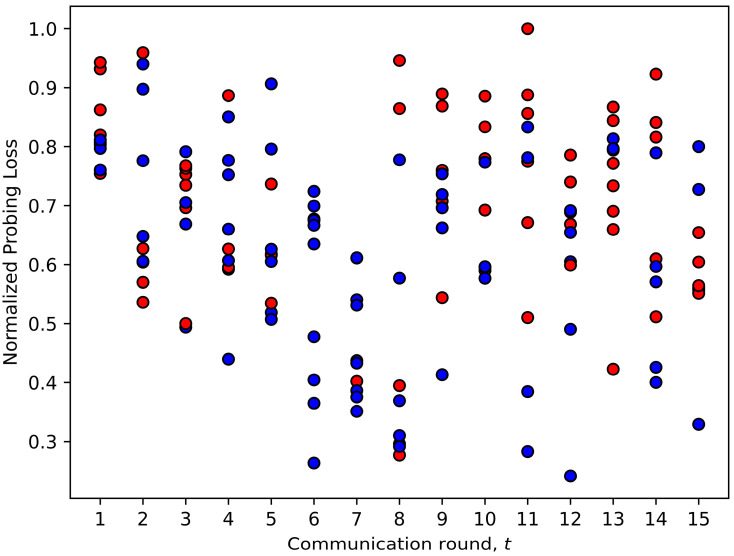
Decisions made by Fiddly based on probing loss.

**Figure 11 sensors-23-02494-f011:**
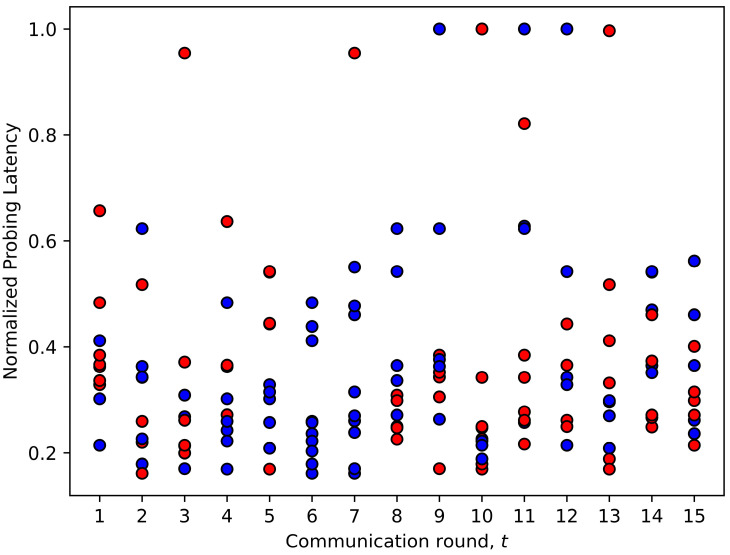
Decisions made by FedDdrl based on probing latency.

**Figure 12 sensors-23-02494-f012:**
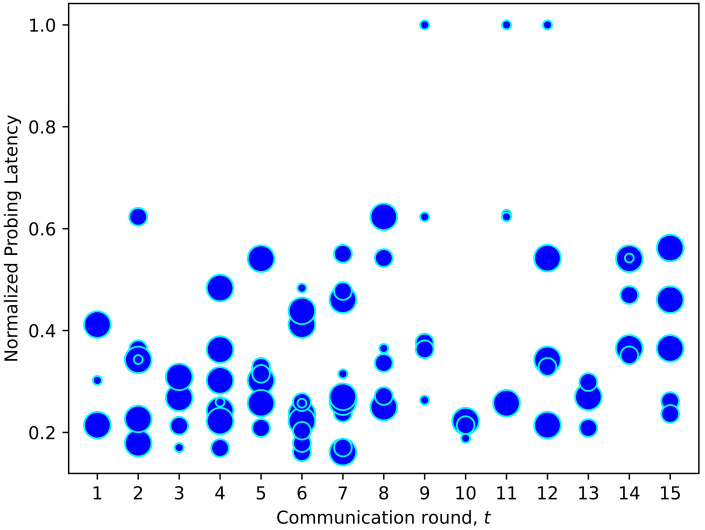
Local epoch count for each selected client in Figure 10. Clients with high probing latency were assigned smaller epoch counts so that the clients could finish local training earlier.

**Figure 13 sensors-23-02494-f013:**
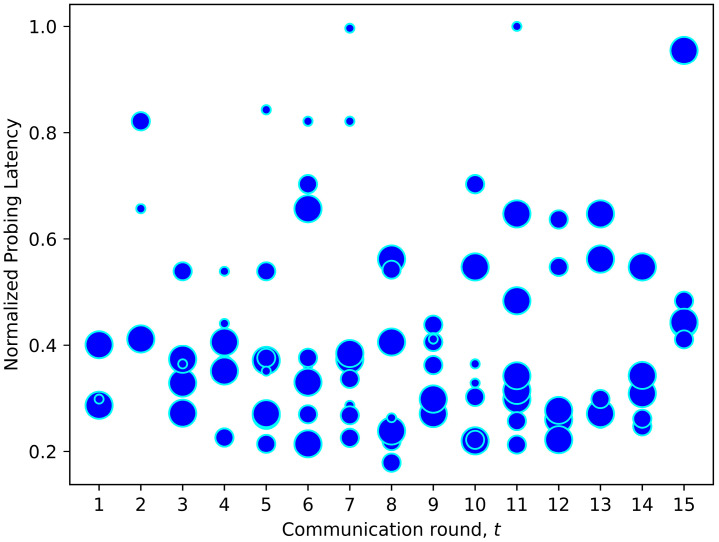
Another example of local epoch adjustment strategy learned by FedDdrl. Selected clients with high probing latency were occasionally assigned large epoch counts to assist the FL convergence.

**Table 1 sensors-23-02494-t001:** Features of existing FL and DRL-based FL algorithms.

Method	Resource Optimization	Client Selection	Local Epoch Adjustment
FedAvg [4]	-	Random	Fixed
FedProx [23]	-	Random	Fixed
FedNova [17]	Computing	Random	Flexible
FAVOR [21]	Computing	DRL Agent	Fixed
TP-DDPG [9]	Computing + Communication	DRL Agent	Fixed
Research work [10]	Computing + Communication	DRL Agent	Fixed
FedMarl [14]	Computing + Communication	DRL Agent	Fixed
Research work [19]	Computing + Communication	DRL Agent	Fixed
Research work [20]	Computing + Communication	Random	Fixed
Research work [29]	Communication	Random	Fixed
Proposed FedDdrl	Computing + Communication	DRL Agent	DRL Agent

**Table 2 sensors-23-02494-t002:** List of key variables defined in the system model.

Notation	Definition
t	Index of communication round
K	The total number of client devices (IoT devices)
N	The total number of client devices selected at each communication round
n	Index of selected IoT devices at communication round t
$H_{t,n}^{b}$	Model broadcasting latency from server to client n
$H_{t,n}^{p}$	Probing training latency for client n
$H_{t,n}^{m}$	Metadata uploading latency from client n to server
$H_{t,n}^{u}$	Model uploading latency from client n to server
$H_{t}$	Complete training latency for communication round t
$B_{n}^{t}$	Communication cost of client n
$B_{t}$	Total communication cost for communication round t
$A_{t}$	Accuracy of the global model at communication round t
$\Delta A_{t}$	Global model’s accuracy improvement
$\phi_{t}$	Client selection matrix at communication round t
$E_{t}$	Local epoch count matrix at communication round t

**Table 3 sensors-23-02494-t003:** List of hyperparameters.

Parameters	Values
Number of agents in each VDN network, N	10
Total number of clients, K	100
Local training dataset distribution, σ	0.8
Learning rate for VDN network	1 × 10^−3^
Target network update interval	5
Number of episodes, $N_{ep}$	40
Number of clients selected for training in each round, C	10
Default number of local epochs (before adjustment by FedDdrl), $E_{t}$	5
Number of communication rounds, T	15
Batch size to update VDN agents, $N_{b}$	32
Initial ε-greedy exploration value	1
Final ε-greedy exploration value	0.1
Replay memory size	300
VDN 1 agent (MLP) size	10 × 256 × 256 × 2
VDN 2 agent (MLP) size	10 × 256 × 256 × 3

**Table 4 sensors-23-02494-t004:** Model accuracy for each FL setting. Bolded indicates the best score, while underlined indicates the second-best score.

Method		MNIST $\left(K=100\right)$	CIFAR-10 $\left(K=100\right)$	CrisisIBD $\left(K=98\right)$
FedAvg		94.6% ± 2.1%	72.8% ± 3.9%	43.2% ± 5.5%
FedAvg with Balanced-MixUp		93.2% ± 2.0%	76.5% ± 1.7%	60.2% ± 1.5%
FedProx (μ = 0.01)		95.6% ± 0.5%	74.5% ± 0.2%	48.1% ± 2.9%
FedProx (μ = 0.01) with Balanced-MixUp		95.4% ± 0.7%	77.8% ± 0.5%	60.7% ± 2.0%
A:	FedMarl ($w_{1}$=1.0, $w_{2}$=0.1, $w_{3}$=0.2)	91.5% ± 1.1%	65.5% ± 2.3%	42.4% ± 3.6%
B:	A + Optimized ($w_{1}$=2.9, $w_{2}$=0.1, $w_{3}$=0.2)	93.2% ± 1.4%	71.7% ± 2.9%	44.4% ± 3.9%
C:	B + Balanced-MixUp	93.3% ± 1.2%	75.0% ± 2.6%	63.3% ± 2.0%
D:	C + Local Epoch Adjustment (FedDdrl)	94.9% ± 1.1%	**78.2% ± 2.4%**	**64.2% ± 1.4%**

**Table 5 sensors-23-02494-t005:** Performance of FedAvg with and without Balanced-MixUp.

Method		MNIST $\left(K=100\right)$	CIFAR-10 $\left(K=100\right)$	CrisisIBD $\left(K=98\right)$
C=5	FedAvg	78.9% ± 9.3%	62.7% ± 2.9%	42.0% ± 3.4%
	FedAvg with Balanced-MixUp	88.1% ± 3.6%	69.4% ± 2.4%	52.9% ± 4.2%
C=10	FedAvg	94.6% ± 2.1%	72.8% ± 3.9%	43.2% ± 5.5%
	FedAvg with Balanced-MixUp	93.2% ± 2.0%	76.5% ± 1.7%	60.2% ± 1.5%

## Data Availability

Not applicable.

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
