# Peer review of "FedDdrl: Federated Double Deep Reinforcement Learning for Heterogeneous IoT with Adaptive Early Client Termination and Local Epoch Adjustment"

_sensors, 2023, doi:10.3390/s23052494_

Round 1

Reviewer 1 Report

1. The issue on Federated learning (FL) is common to allows multiple devices to collaboratively train a global model without sharing their sensitive and bandwidth-hungry data. You need to explore and provide the problems on the areas of "without sharing their sensitive and bandwidth-hungry data" Please add them on the paper.

2. Please provide detail of common result on the FL trends and problems.

3. Please show and provide in detail with the detail of background problems as well as the resilt of an efficient and robust FL framework.

4. Please provid in detail with the result and critize the process and result of  maximise the model’s accuracy while minimising training latency and communication costs.

5.  Please provide in detail with the process on how to determine the Heuristic solutions  in maximizing all three objectives simultaneously.

6. We can not find the role and function of  deep reinforcement learning (DRL) based methods on your paper. Please add them on the paper

7. We can not find on the implications on the a client selection policy or early client termination to select clients with higher computing and network resources.

8. Please provide in detail with the problems on the , existing DRL-based FL. Please compare them with the that  not fully utilize the computing resources of clients.

9. Please add on the paper with the industries persfective resul.

Author Response

Dear reviewer, 

Best regards,

Yi Jie Wong et al. 

Reviewer 2 Report

The paper very clearly augmented the need for this proposed technique with detailed literature. The major drawback in the previous studies was not fully utilize the computing resources of clients by adjust ing the local epoch. Therefor authors proposed FedDdrl, a FL framework that exploits double deep reinforcement learning (DDRL) to jointly perform adaptive early client termination and dynamic adjustment of local epoch.

To improve the overall quality of paper authors need to address following:

1. The contribution of the algorithm is ambiguous. FedDdrl algorithm needs more clarity in terms of some comments.

2. All of the abbreviations should be provided with the full definition at their first occurrence, and refrain from repeating the use of the full form after the first time.

3. I strongly advise authors to include a comparison table in their related work. For the sake of comparison, newly published papers should be used.

4. Normally, authors never start the heading after healing without explaining the first one like in this paper, Proposed method, Related  needs some text then we can start Federated Learning and Deep Reinforcement Learning.

5. The authors did not specify which simulation platform was used for the work.

6.  The proposed algorithm was tested in a simulation-based environment. I firmly believe that there is huge difference between a real-time environment and a simulation-based. Therefore, I highly suggested authors perform this algorithm test on an actual environment.

Author Response

(The authors gave the same response as above.)

Reviewer 3 Report

This is interesting research. The authors discussed the issue of how the data that emerged from IoT devices uploaded to centralized servers consume bandwidth-hungry and require a high communication cost. Although Federated learning (FL) has emerged as one of the promising solutions, still some performance issues appear.

The objectives of this research are to maximize the global model’s accuracy while minimizing the FL system's training latency and communication cost. They used Markov Decision as well as adopted two DRLs based on Value Decomposition Networks (VDNs) as the policy networks. To speed up convergence speed, they also adopt the recently proposed balanced-MixUp

Author Response

(The authors gave the same response as above.)

Round 2

Reviewer 1 Report

1.  We suggest you explore more detail the issues problems on  the Federated learning (FL).

2. How do you solve the issues that related with  multiple devices to collaboratively train a global model without sharing their sensitive and bandwidth-hungry data.

3. Please show in detail with detail  efficient and robust FL framework that maximise the model’s accuracy

4. Please show in detail with startefies  minimising training latency and communication costs.

5. Please provide in detail with solutions of  solve  difficulties in maximizing all three objectives simultaneously.

6. How do you provide in detail with stages of  deep reinforcement learning (DRL) based methods.

7. How you provide in detail with strategies iof  a client selection policy or early client termination to select clients with higher computing and network resources.

8. Please provide the correctness of  FedDdrl, a FL framework that exploits double deep reinforcement learning (DDRL).

Author Response

Hi Reviewer 1,

Best regards,

Wong Yi Jie

Round 3

Reviewer 1 Report

The revised paper is ok to be published